# Mechanochemistry and Eco-Bases for Sustainable Michael Addition Reactions

**DOI:** 10.3390/molecules27103306

**Published:** 2022-05-21

**Authors:** Yvette Lock Toy Ki, Armelle Garcia, Franck Pelissier, Tomasz K. Olszewski, Alicja Babst-Kostecka, Yves-Marie Legrand, Claude Grison

**Affiliations:** 1Laboratory of Bio-inspired Chemistry and Ecological Innovations (ChimEco), UMR 5021, CNRS, University of Montpellier, Cap Delta, 1682 rue de la Valsière, 34790 Grabels, France; yvette.lock-toy-ki@cnrs.fr (Y.L.T.K.); armelle.garcia@cnrs.fr (A.G.); franck.pelissier@cnrs.fr (F.P.); yves-marie.legrand@umontpellier.fr (Y.-M.L.); 2Department of Physical and Quantum Chemistry, Faculty of Chemistry, Wroclaw University of Science and Technology, Wybrzeze Wyspianskiego 29, 50-370 Wroclaw, Poland; tomasz.olszewski@pwr.edu.pl; 3Department of Environmental Science, The University of Arizona, 1110 E. South Campus Drive, Saguaro Hall, Building 33, Tucson, AZ 85721, USA; ababstkostecka@arizona.edu

**Keywords:** Michael addition, mechanochemistry, eco-base, alien invasive plants, phytomanagement

## Abstract

The Michael addition reaction was revisited with a full focus on sustainability combined with efficiency, using mechanochemistry in mild conditions. First, the synthesis of cyclopentenone derivatives was chosen as a model reaction to find optimal conditions in mechanochemistry while using classical but weak bases. The reaction was efficient (84–95% yields), fast (2–6 h), solvent free, and required 0.1 equivalent of base. Aiming to reach greener conditions, classical bases were then replaced using new bio-sourced bases, called Eco-bases, that were easily prepared from plants and led to heterogeneous catalysts. The composition and structure of Eco-bases were characterized by MP-AES, XRPD, EBSD/EDS, HRTEM/EDX and ion chromatography. Interestingly, a high ratio of potassium was observed with the presence of K_2_Ca(CO_3_)_2_ for the most effective Eco-base. The new Eco-bases were used for the mechanical-assisted construction of functionalized alkenone derivatives. The versatility of the method has been successfully applied with good to excellent yields to different Michael donors and acceptors. Eco-bases were recycled and reused four times with the same performances. Combining Eco-bases and mechanochemistry in Michael addition reactions allowed reaching a maximum degree of sustainability (efficient, rapid, low catalyst loading, solvent-free reactions with bio-sourced catalysts) and participating in the development of mechanochemistry in sustainable chemistry.

## 1. Introduction

Michael addition is an interesting methodology for the building of C–C and C–X links. With the possible variety of Michael donors and acceptors, the reaction is a versatile synthetic tool for constructing simple and complex molecules. Michael and heteroatom-Michael additions are widely used for the production of drugs, perfumes, agrochemical products and polymers with numerous applications in emerging technologies including biomedical, pharmaceutical, optoelectronic composites, adhesives, and coatings [1]. Michael addition also presents a number of advantages in Green Chemistry. Its mechanism displays a 100% atom economy. Michael addition involves an addition to carbon–carbon double bonds containing an electron-withdrawing group in mild conditions. The reaction has usually been performed in protic and ecofriendly solvents, such as alcohols, with a weak base (tertiary amines, alkali hydroxides, guanidines, amidines, phenolates, fluorides etc.) [2,3]. If intramolecular Michael additions are performed with sub-stoichiometric or catalytic bases in a homogenous medium [4], the bases are rarely recyclable [5,6,7].

Many efforts have been carried out to design heterogeneous catalysts to promote base reuse and recycling. Two strategies have been developed. The first one is based on grafting the base on a solid support (silica, polystyrene) [3]. However, the preparation of these solid catalysts is not consistent with the principles of Green Chemistry. For example, aminopropylsilane can be grafted onto silica using trimethoxyamino propyl silane (TMPS) by heating to 100 °C in toluene. TPMS is prepared by the addition of hydrotrimethoxysilane onto allylamine with a rhodium catalyst (rhodium phosphide or its oligomer) [8]. The life cycle assessment of such a process is clearly not satisfactory. The second strategy is based on using metal-catalyzed reactions, since they present several advantages, as recyclability and easy separation. However, the cost and hazard of metals limit their use in an organometallic Michael addition.

Recently, efforts have been made in using metals for more sustainable Michael addition reactions. For example, microcrystalline cellulose-immobilized nanoparticles with iron (Fe_3_O_4_@MCC) were described to promote Michael addition reactions in an aqueous ethanolic medium and gave high yields in mild conditions [9]. The catalyst was recycled and reused without considerable depreciation in catalytic activity after 5 runs. However, the preparation of the catalyst is questionable in terms of Green Chemistry.

Metal nanoparticles of magnesium oxide have also successfully been used as catalysts in Knoevenagel reactions, while using the Grindstone method, without solvent and at room temperature [3]. Inspired by these last two examples and considering the similarities between the Knoevenagel reaction and the Michael addition, especially the formation and the addition of a carbanion nucleophile from an activated methylene precursor, we have first studied the potential of weak bases in Michael addition using mechanical-assisted methods. Aiming to reach greener conditions, we then prepared and characterized bio-sourced bases, called Eco-bases, and investigated their catalytic efficiency in some mechanical-assisted Michael addition reactions (Figure 1).

## 2. Methods

### 2.1. General Information

Gas Chromatography coupled with Mass Spectroscopy (GC–MS) analyses were performed on a Thermo Scientific™ Trace 1300 GC coupled with an ISQ QD quadrupole. The detection system was connected with a Thermo TG-5SilMS column (0.18 µm × 0.18 µm × 20 m). GC-FID analyses were performed on a similar column connected to a flame ionization detector (FID). In each case, hydrogen was used as a carrier gas (1 mL min^−1^), using the following temperature program: 80 °C isothermal (1 min), 80 to 260 °C gradient at 40 °C min^−1^, then 260 °C isothermal (1 min).

The samples were prepared in ethyl acetate, and biphenyl was used as an internal standard for GC-FID quantifications. Mass spectra were recorded in impact electronic mode at 70 V, and identification was made by the NIST 14 database.

Nuclear Magnetic Resonance (NMR) spectra were recorded on a Brüker Avance III HD 400 MHz at 20 °C using deuterated chloroform (CDCl_3_) as solvent. The ^1^H and ^13^C frequencies were 400 and 100 MHz respectively.

Infrared (IR) spectra were recorded on a Perkin Elmer Spectrum 100 FT-IT, ATR mode, between 700 and 3000 cm^−1^.

Microwave plasma-atomic emission spectrometer (MP-AES) coupled with an SPS4 autosampler from Agilent Technologies™ was used to study the mineral composition of the Eco-bases. The mineral composition of parts of several plants was determined by MP-AES after a heat treatment under air flow at 550 °C during 4 h. MP-AES analyses were performed using the metal analysis of total dissolved solutes in water. The samples were digested in 10 mL of reversed aqua regia (1:2, hydrochloric acid (37%): nitric acid (65%)) under a microwave-assisted digestion (Multiwave-Go Anton Paar) with the following program: 20 to 165 °C in 20 min and then 10 min isothermal at 165 °C. Samples were filtered and then diluted to 0.4 mg L^−1^ in 1% aqueous nitric acid. Mineral compositions were determined by using a microwave plasma-atomic emission spectroscopy (MP-AES) 4200 (Agilent Technologies, Santa Clara, CA, USA) equipped with a concentric nebulizer and a double-pass cyclonic spray chamber. The pump speed during analysis was kept at 10 rpm, and the sample introduction tube diameter was 0.89 mm. The analytical cycle consisted of 30 s rinsing with aq. 1% nitric acid followed by 25 s of sample uptake (pump speed 40 rpm) and then 20 s of equilibration before the reading at preselected integration times (pump speed 10 rpm). The integration time was set to 3 s for all elements. Unless otherwise stated, the automatic background correction mode available in the software was used. All analysis results were performed in triplicate.

X-Ray Powder Diffraction (XRPD) data measurements on samples dried at 100 °C for 2 h were conducted using a BRUKER diffractometer (D8 Advance, with Cu Kα radiation at 1.54086 Å) equipped with a LynxEye detector. Diffraction patterns were analyzed with DIFFRAC-EVA software and several XRD databases (Crystallography Open Data Base 2016, and PDF 2011, 2017 or 2018).

Electron backscattered diffraction (EBSD) and energy-dispersive spectroscopy (EDS) analysis were performed in a scanning electron microscope (CamScan X500FE CrystalProbe) at Geosciences Montpellier (France). EBSD maps with step sizes ranging from 0.2 to 0.7 µm and forward scatter electron (FSE) images with a resolution up to 5 nÅ were acquired on small areas within each crystal using a working distance of 25 mm, a tilt angle of 70°, an acceleration voltage of 20 V, and a beam current of 10 nA. EBSD data were acquired and treated with the AZtecSynergy software from Oxford Instruments HKL.

High-resolution transmission electron microscopy (HRTEM) analyses were performed at 200 kV on a JEOL 2200 FS equipped with a CCD Gatan Ultrascan 4000 CCD (4092 × 4092 px2) at the MEA platform (University of Montpellier, France). Scanning transmission electron microscopy-energy dispersive X-ray spectroscopy (STEM-EDX) was used to obtain chemical mapping with a probe size of 0.7 nm. The EDX spectra were recorded on a Silicon Drift X-MAX^n^ 100TLE detector from Oxford Instrument. Samples were embedded in LR White resin and cut (thin section of 70 nm) with a Leica UC7 ultramicrotome equipped with a Diatome diamond. Sections were deposited on a carbon-coated copper grid (300 mesh).

Ionic chromatography analyses were performed at the Laboratory of Physical Chemistry and Microbiology for Materials and Environment (LCPME-UMR 7564 CNRS-University of Lorraine) with 882 Compact IC from Metrohm equipped with a chemical suppressor and a conductimetric detector. A Metrosep A Supp 16 – 250/4.0 column with a guard column and RP3 pre-column thermostated at 55 °C were used. A NaOH (25 mM) solution with a soda lime guard is used as eluent (flow rate: 0.8 mL min^−1^).

Mechanical-assisted Michael addition reactions were performed with a Planetary Ball Mill Machine referenced Retsch PM 100. The reaction mixture was ground for a specific time (see tables) with a rotation speed of 500 rpm (8.33 Hz) with a pause interval of 10 min every hour.

### 2.2. General Procedures

#### 2.2.1. Harvest and Preparation of the Eco-bases

*Arabidopsis halleri* (Ah) leaves have been collected in a heavily industrialized region of Poland in the vicinity of the Zn smelter of the Bolesław Mine and Metallurgical Plant near the city of Olkusz.

*Nocceae caerulescens* (Nc) and *Anhyllis vulneraria* (Av) leaves were harvested before flowering in Les Avinières, a former Zinc mining site in Saint-Laurent-le-Minier.

*Arabidopsis halleri* (Ah), *Nocceae caerulescens* (Nc) *and Anhyllis vulneraria* (Av) are well-studied hyperaccumulators of Zn, which were widely used in phytoremediation programs. The population from the investigated location was previously shown to contain extremely high and consistent concentrations of Zn in aerial parts.

*Fallopia japonica* (Fj) is another alien invasive plant in European wetlands. Leaves have been collected in Saint-Bauzille-de-Putois (Hérault) and Thoiras (Gard), France, in a context of integrated wetland management by ETPB Gardons and Grison’s team. This is a highly invasive and widespread alien species in Europe, and it is also listed as one of the most potent global invasive plant species.

*Salix alba* (Sa) leaves were harvested in Laon, Grand-Est region, France. *Salix alba* is a common tree species, which is native to the wetlands of the northern hemisphere, especially Europe, temperate Asia and North Africa. Its biomass is very abundant.

Oyster shells were given by oyster aqua-culturists from Thau lake in the Mediterranean area, France.

Leaves were first washed twice in deionized water and oven-dried at 80 °C. The obtained solid (30 g) was then thermally treated in an oven under air flow. The temperature program consisted of a first gradient from 25 to 350 °C in 1 h then 2 h at 350 °C, which was followed by a second gradient from 350 to 550 °C in 1 h then 4 h at 550 °C before a slow cooling in the oven to obtain Eco-base (5 g).

#### 2.2.2. Procedures for Dimethyl-2-(3-oxocyclopentyl) Malonate **3**

Weak-bases (1 equiv of Zn for ZnO, 1 equiv of Ca for CaO, EcoCaOx, CaCO_3_, 1 equiv of Mg for MgO, 0.08 equiv of K for K_2_CO_3_, KHCO_3_, K_2_CO_3_ + CaO_,_ K_2_CO_3_ + CaCO_3_), dimethyl malonate (580 µL, 5 mmol, 1 equiv), 2-cyclopenten-1-one (420 µL, 5 mmol, 1 equiv) were added to a ball mill reactor with 100 balls of 5 mm diameter. The reactor was grinding for 2 h at 8.33 Hz. Every hour, the rotation was stopped for 10 min, and the rotation direction was reversed. The ball mill reactor was washed three times with EtOAc (25 mL). The suspension was filtered with ethyl acetate (25 mL), and the filtrate was concentrated under vacuum.

Eco-base (5.58 mmol, 1.1 equiv of K), dimethyl malonate (580 µL, 5 mmol, 1 equiv), and 2-cyclopenten-1-one (420 µL, 5 mmol, 1 equiv) were added to a ball mill reactor with 100 balls of 5 mm diameter. The reactor was grinding for 2 h at 8.33 Hz. Every hour, the rotation was stopped for 10 min and the rotation direction was reversed. The ball mill reactor was washed three times with EtOAc (20 mL). The suspension was filtered with ethyl acetate (25 mL), and the filtrate was concentrated under vacuum.

Dimethyl 2-(3-oxocyclopentyl) malonate **3** was purified by distillation (Eb_150 mbar_ = 64–65 °C).

Dimethyl 2-(3-oxocyclohexyl)malonate and dimethyl 2-(2-nitro-1-phenylethyl)malonate were purified by flash chromatography (cyclohexane: ethyl acetate, 9:1).

## 3. Results and Discussion

### 3.1. Mechanochemical Michael Addition Promoted by Weak Bases

First, we studied the mechanochemical Michael addition of dimethylmalonate **2** onto 2-alkyl cyclopentenones **1**. This 1,4-addition is the key step of the industrial synthesis of perfume ingredients such as methyl dihydrojasmonate [9,10,11]. Considering the recent performances of metal oxides in Knoevenagel reactions and Michael addition reactions, several oxides and other mineral bases of similar basicity were tested. The reaction was performed without solvent in mild conditions (Table 1).

The reaction was not possible by direct mechanocatalysis in the absence of base (Table 1, entry 1). The use of metal oxides (MgO, ZnO) combined to mechanical energy, which were successfully used in Knoevenagel reactions [3], did not promote the Michael addition (Table 1, entries 2–3). The use of slightly stronger bases such as CaO and Eco-CaOx, derived from oyster shells [12], led to a total conversion of **1** (Table 1, entries 5–6). While CaCO_3_ and KHCO_3_ were ineffective, K_2_CO_3_, K_2_CO_3_ + CaO and K_2_CO_3_ + CaCO_3_ exhibited excellent activity (Table 1, entries 4, 7–10). The comparison between batch mode and mechanochemical activation was interesting (Table 1, entries 7–10). When the substrate was bulky (R = C_5_H_11_), the presence of the base was not sufficient. The mechanical activation was required (Table 1, entries 9, 10).

In conclusion, mechanochemistry can be adapted to Michael additions. Different advantages can be claimed: (i) the reaction was very effective, even with bulky substrates; (ii) the reaction conditions were mild: the reaction was fast (2–6 h) at room temperature, so the conditions were energy-efficient; (iii) the reaction was solvent free and prevented waste production; (iv) the use of weak bases such as K_2_CO_3_ was sufficient. Mechanochemistry can therefore contribute to developing green and sustainable Michael addition reactions. Additionally, in the aim to increase the sustainability of the reaction, we chose to substitute K_2_CO_3_ by bio-sourced bases, called Eco-bases. Indeed, the industrial manufacture of K_2_CO_3_ is based on the carbonation of KOH, which is produced by, not so green, electrolytical mercury processes [13]. Progresses in terms of life cycle assessments are clearly needed.

### 3.2. Mechanochemical Michael Addition Promoted by Eco-Bases

The Grison group has shown that the remediation phytotechnologies, such as phytoextraction [14,15,16,17,18,19,20,21,22,23,24,25,26,27], rhizofiltration [28,29,30,31] and biosorption [32,33,34,35,36] generate biomass which can be turned into innovative ecocatalysts. Among the plant species that can be transformed, here, we selected the ones that were the richest in potassium (Figure 1).

First, we have used an Zn-hyperaccumulating plant, *Anthyllis vulneraria* (Av), *Noccaea caerulescens* (Nc) and *Arabidopsis halleri* (Ah). The first two have been harvested in Les Avinières, which is a former zinc-mining site in Saint-Laurent-Le-Minier, southern France. The second one has been harvested in a mine and metallurgical site in Poland (Ah). Then, we have studied the used of an alien invasive species in Europe, *Fallopia japonica* (Fj). This plant species is very abundant, and its richness in Mg and K make it interesting to design an Eco-base. Finally, a common and available woody tree, *Salix alba* (Sa), has been tested for comparison.

Considering the mineral chemodiversity of these plants, a comparative study of chemical composition of Eco-bases was performed in order to identify the best candidates for mechanochemical Michael addition.

#### 3.2.1. Preparation of Eco-bases

For Eco-bases from Zn-hyperaccumulators and invasive species, bulk samples of leaves were collected from  ≈1000 individual plants within a sample area of ≈0.1 km^2^. Sampling at the metalliferous sites was conducted across an area that is consistently affected by the source of pollution, and the harvested plant biomass was thoroughly homogenized to minimize biases from site heterogeneity. As for the reference plant material *S. alba*, leaves were collected from four trees, from an area of 2 km^2^.

The dried leaves, from each plant species, were thermally treated under air flow at 550 °C for 4 h, in order to remove the organic matter.

#### 3.2.2. Characterization of Eco-bases

The elemental composition of the Eco-bases was determined by MP-AES analyses (Table 2).

As expected, Eco-base-Av, Eco-base-Nc and Eco-base-Ah exhibited a high rate in Zn (5.02−10.16%) and K (10.46−16.99%). While Eco-base-Ah was the richest in Zn (10.16%)*,* Eco-base-Fj had the highest rate in K (20.13%) and Mg (6.21%). The mineral composition of Eco-base-*Sa* was more usual (Table 2, entry 5, [37]). Accordingly, Eco-base-Ah and Eco-base-F_j_ were selected for further steps due to their most promising mineral composition: the highest concentrations of Mg, very high rates of K, and the highest Zn rate in Eco-base-Ah.

X-Ray Powder Diffraction (XRPD) analyses were then performed to characterize the crystalline complexes found in the two Eco-bases that exhibited the most interesting composition, Eco-base-Ah and Eco-base-Fj (Table 3).

The structures of Eco–base-Ah showed an unusual mixture of zinc silicate, such as K_2_ZnSiO_4_ and ZnSiO_3_, an aluminic species with interesting properties, Paragonite [38], and usual potassium species, KCl and K_2_SO_4_ (Table 3, entries 1, 2).

Mixed metal oxides as (Mg_0.9_Fe_0.1_)O and (Mg_0.9_Zn_0.1_)O were observed in Eco–base-Ah (Table 3, entry 1). The formation of magnesium oxide was surprising at 550 °C compared to standard temperature conditions (800 °C). Interestingly, the presence of mixed magnesium-zinc oxide could enhance the reactivity of MgO [21,22,25,29]. With MgO being ineffective in this reaction (see Table 1), it was interesting to examine the effect of ZnO on the reactivity of MgO.

Eco-base-Fj exhibited a carbonate of potassium and calcium: K_2_Ca(CO_3_)_2_ (Table 3, entry 2). The formation of this mixed carbonate is in agreement with the temperature of the thermal treatment [37]. In the preliminary study (Table 1), we have showed that only K_2_CO_3_ was efficient to promote Michael addition. The reactivity of K_2_Ca(CO_3_)_2_ was still unknown in this reaction.

Complementary assessments were performed to identify other basic entities. The combination of chemical data from energy dispersive X-ray spectrometry (EDS) and crystallographic data from electron backscattered diffraction (EBSD) has allowed us to identify oxyapatite calcium phosphate for Eco-base-Fj (see Appendix A). Although the temperature of thermal treatment was below 600 °C, the formation of oxyapatite calcium phosphate was consistent with Hedayati’s studies [39,40], which described the presence of K_2_Ca(CO_3_)_2_ and (CaPO_4_)_6_ CaO in the ashes of poplar. This mineral species has been described as a basic catalytic material [41].

The morphology of Eco-bases was analyzed by HR-TEM and the structural distribution of mineral species was mapped by STEM-EDX for Eco-base-Ah (Figure 2) and Eco-base-Fj (Figure 3).

HR-TEM images of the Eco-bases clearly confirmed their granular morphology with a particle size in the range of 50 to 200 nm. Round particles of about 10–50 nm of diameter seemed to blend together into a mineral matrix. STEM-EDX confirmed that potassium, calcium, magnesium and phosphorus were present, as expected based on the MP-AES analyses. The polymetallic structure of Eco-bases was supported by the aggregation of potassium, oxygen and all the other elements in round small particles.

The co-localization of Mg and Zn confirmed the formation of (Mg_0.9_Zn_0.1_)O for Eco–base-Ah. Likewise, the observation of mixed carbonate of K_2_Ca(CO_3_)_2_ was consistent with the co-localization of Ca and K for Eco–base-Fj. The simultaneous presence of Ca, S and O for Eco-base-Ah. *Arabidopsis halleri* belongs to the Brassicaceae family, which can be explained by the presence of CaSO_4_.

#### 3.2.3. Eco-bases Reactivity in the Mechanochemical Michael Addition

##### Synthesis of Dimethyl 2-(3-oxocyclopentyl) Malonate **3**

We studied the mechanochemical reactivity of Eco-bases in the Michael addition reaction between dimethylmalonate **2** and 2-alkyl cyclopente-2-enones **1** for comparative purposes with the conventional bases, which were previously used (Table 4).

It should be noted that the batch mode was not adapted to the Michael addition catalyzed by Eco-bases. With one equivalent of dimethylmalonate **2**, the reaction failed (Table 4, entry 1). Some difficulties in mechanical agitation required the use of an excess of reagent **2** (Table 4, entry 2).

The mechanochemical reactions proceeded faster than batch and did not require an excess of one reagent (Table 4, entries 3–6). Excellent yields were obtained with all Eco-bases (81–97%). Moreover, one hindered Michael acceptor (R_1_ = C_5_H_11_) could be converted to **3** adduct with the combination of Eco-base-Fj and mechanochemistry (Table 4, entry 10). While Eco-base-Ah and Eco-base-Sa required a long reaction time (24 h and 16 h respectively, entries 8−9), a similar conversion was obtained in only 2 h with Eco-base-Fj. Finally, the desired product was obtained with a high conversion (84%) in 6 h with Eco-base-Fj.

The structures established by XRDP can explain the difference of reactivity between Eco-bases (Table 3). The highest performance of Eco-base-Fj could be due to the presence of K_2_Ca(CO_3_)_2_ as a basic entity. This hypothesis is consistent with the results of Table 1, which have shown the high reactivity of potassium carbonate in mechanochemical Michael addition.

Supplementary experiments were performed to calculate the amount of K_2_Ca(CO_3_)_2_ in Eco-base-Fj. Indeed, as shown in Table 3, three potassium species were formed: KCl, K_2_SO_4_ and K_2_Ca(CO_3_)_2_. We evaluated the amount of each potassium species by dissolution in water and ion chromatographic analysis of the aqueous layer. The conditions and results are presented as Appendix A. An average of 18% of carbonate was found versus 62% of chloride and 20% of sulfate. We can, therefore, conclude that only 0.08 equivalent of potassium active species promoted the mechanochemical Michael addition. This result is remarkable compared to standard catalysts, which usually require to be used in excess.

###### Recyclability of Eco-base-Fj

A major challenge of green catalysis is the ability to recycle and to reuse the catalyst. In order to test the potential for recycling and reuse of our best Eco-base, Eco-base-Fj was washed with ethyl acetate, dried (100 °C, 5 h) and used in a new run to yield the desired product. Eco-base-Fj could be used for up to four cycles, without any decrease in activity. These results thus illustrate the stability of the Eco-base (Table 5).

###### Scope of Michael Addition Promoted by Eco-base and Mechanochemistry

Subsequently, the methodology was applied to a selected spectrum of Michael acceptors (Table 6) and to Michael donors (Table 7) to test the scope of the presented protocol.

The Eco-base-Fj was found to efficiently mediate the reaction when using cyclohex-2-en-1-one (Table 6, entry 2) or *trans*-chalcone (Table 6, entry 3), and dimethylmalonate. It should be noted that the conversions were excellent even with chalcone, which was the least reactive Michael acceptor. In the case of nitrostyrene, the yield was poor (entry 4), and some degradation occurred. The reaction failed with cinnamonitrile and methyl cinnamate. The screening of Michael acceptors revealed that the methodology was especially well adapted to enones.

The use of β-dicarbonyl compounds (Table 7, entries 1, 2) or ethyl acetoacetate (Table 7, entry 3) instead of dimethylmalonate led to the expected Michael adducts when the reaction was performed at room temperature for 2 h (Table 7). Cyclohexane-1,3-dione and ethyl acetoacetate gave the best conversions (75 and 95%, respectively). Surprisingly, 2,4-pentandione was less reactive.

## 4. Conclusions

Mechanochemistry and Eco-bases are an innovative combination to develop greener and sustainable Michael addition. This approach offers numerous advantages: faster reaction, mild and solvent-free reaction conditions, efficiency with only 0.01 equivalent of potassium active species, resource and energy economy, catalytic conditions, recyclability, bio-sourced and available bases. It demonstrates the remarkable practical utility and paves the way toward sustainable and greener catalytic chemistry. In addition to the advantages of sustainable chemistry, this approach provides other major environmental benefits: (i) restoration of degraded land ecosystems by the phytomanagement of contaminated soils and valorization of contaminated biomass in organic synthesis; (ii) use of invasive plant species as starting materials to produce Eco-bases. It is an opportunity for the recovery of alien invasive plants of wetlands such as *Fallopia japonica*, and it provides interesting support for the control of their proliferation. This new generation of ecocatalysts exhibits environmental and scientific benefits, suggesting a promising life cycle analysis (Figure 4).

## Data Availability

Not applicable.

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
