# Peer review of "Mechanochemistry and Eco-Bases for Sustainable Michael Addition Reactions"

_molecules, 2022, doi:10.3390/molecules27103306_

Round 1

Reviewer 1 Report

The manuscript submitted by Grison and coworkers describes the use of Eco-bases as a promoter of Michel additions.

The results presented in this manuscript are fascinating and may deserve publication in Molecules after some significant changes.

  1. Introduction. Please remember that Michael additions are broad scope transformations pretending to build a new C-X and not only C-C bonds. Oxa-Michael, Aza-Michael, and thia-Michael, among others, are very useful and well-known chemical transformations. So, a short mention of those complementary strategies should be included in the introduction.
  2. Introduction. I agree with the authors concerning the previous catalytic approaches of Michael additions. However, there are plenty of examples where intramolecular Michael additions are performed with sub-stoichiometric or catalytic bases in a homogenous medium. Please check at least the work of David Evans, Joëlle Prunet and George O’Doherty, a recent review covering this topic is Synthesis 2018, 50, 3997.
  3. Experimental section: Only descriptions for Eco-Ah and Eco-Fj are presented. However, Eco-CaOx, Eco-Av, Nc, and Sa were used. Please provide a short description of the used material.
  4. Conclusions: In the text, page 10, paragraph 1, the authors claimed that only 0.08 equiv. of potassium active species promote the reaction. However, they used 1.1 equivalent to study the reaction scope, and they concluded that only 0.01 equiv is necessary. So, please clarify this point and provide convenient evidence of the catalytic role of the Eco-base.

Further criticism,

The tentative composition of the Eco-bases is provided. The authors suggested that potassium active species (bases) are responsible for the reaction, and unfortunately, there is no evidence of this fact. The reaction under traditional conditions always used 1 equiv of the base. I strongly suggest a couple of experiments to clarify this point. A) an experiment using catalytic amounts of potassium bases, b) an experiment trying to mimic the composition of an Eco-base to determine if there is a synergic effect with other uncharacterized compounds in the Eco-base.

Finally, the authors claimed that the Eco-base could be recovered and reused. However, the drying process includes heating at 500ºC. How are they sure that the base is really reused and that the drying process is not a re-activation process?

Can the authors provide a comparative experiment using the base after a regular drying (evaporation of ethyl acetate and 100ºC heating)?

After this revision, I will be happy to support the publication of this papper in Molecules.

Author Response

Reviewer 1

The manuscript submitted by Grison and coworkers describes the use of Eco-bases as a promoter of Michel additions.

The results presented in this manuscript are fascinating and may deserve publication in Molecules after some significant changes.

  1. Introduction. Please remember that Michael additions are broad scope transformations pretending to build a new C-X and not only C-C bonds. Oxa-Michael, Aza-Michael, and thia-Michael, among others, are very useful and well-known chemical transformations. So, a short mention of those complementary strategies should be included in the introduction.

Precisions about the synthetic potential of Michael additions have been added in introduction.

Michael addition is an interesting methodology for the building of C-C and C-X links. With the possible variety of donors and acceptors of Michael, the reaction is a versatile synthetic tool for constructing simple and complex molecules. Michael and heteroatom-Michael additions, are widely used…”

  1. Introduction. I agree with the authors concerning the previous catalytic approaches of Michael additions. However, there are plenty of examples where intramolecular Michael additions are performed with sub-stoichiometric or catalytic bases in a homogenous medium. Please check at least the work of David Evans, Joëlle Prunet and George O’Doherty, a recent review covering this topic is Synthesis 2018, 50, 3997.

            The sentence has been modified:

            “If intramolecular Michael additions are performed with sub-stoichiometric or catalytic    bases in a homogenous medium (Synthesis 2018, 50, 3997), the bases are rarely         recyclable”

  1. Experimental section: Only descriptions for Eco-Ah and Eco-Fj are presented. However, Eco-CaOx, Eco-Av, Nc, and Sa were used. Please provide a short description of the used material.

            Experimental precisions have been added in the experimental part.

  1. Conclusions: In the text, page 10, paragraph 1, the authors claimed that only 0.08 equiv. of potassium active species promote the reaction. However, they used 1.1 equivalent to study the reaction scope, and they concluded that only 0.01 equiv is necessary. So, please clarify this point and provide convenient evidence of the catalytic role of the Eco-base.

            0.08 equiv. of potassium active species promote the reaction; it is correct. However,      0.01 equiv.is a mistake. This has been corrected.

    As illustrated by table 3, 1.1 equivalent is the sum of three potassium species: KCl,        K2SO4 and K2Ca(CO3)2. We evaluated the amount of each potassium species by           dissolution in water and mechanochemical Michael addition. This result is remarkable compared to standard    catalysts which usually require to be used in excess.

Further criticism,

The tentative composition of the Eco-bases is provided. The authors suggested that potassium active species (bases) are responsible for the reaction, and unfortunately, there is no evidence of this fact. The reaction under traditional conditions always used 1 equiv of the base. I strongly suggest a couple of experiments to clarify this point. A) an experiment using catalytic amounts of potassium bases, b) an experiment trying to mimic the composition of an Eco-base to determine if there is a synergic effect with other uncharacterized compounds in the Eco-base.

We agree with this comment. Different experiments have been carried out to prove the role of potassium species:

  • As indicated in table 1, K2CO3 (0.08 equiv) is an excellent catalyst of the reaction (entries 7 and 8), even with a hindered substrate (entry 11);
  • We tried to prepare the supposed active species (K2Ca(CO3)2, Fairchildite

 Numerous methods have been described in the literature (see references below). The following table summarizes the conditions of preparation:

Entry

Starting material

Conditions

Product

Reference

1

K2CO3, CaCO3

500 °C, 96 h

K2Ca(CO3)2

[1]

2

K2CO3, CaCO3

600 °C, 72 h

K2Ca(CO3)2

[1]

3

K2CO3, CaCO3

Cooling from 830 to 650 °C for 30 min

K2Ca(CO3)2

[1]

4

K2CO3, CaCO3

505 °C, 8 jours

Buetschliite

[2]

5

K2CO3, CaCO3

585 °C, 4 jours

Fairchildite

[2]

6

KHCO3sat, CaCO3

66 °C, 5 jours

K2Ca(CO3)2

[2]

7

K2CO3sat, CaCO3

55 °C, 6 jours

K2Ca(CO3)2, CaCO3

[2]

8

K2CO3, CaCO3 (80 mol%)

50 bar under CO2

 [3]

These bibliographical data can lead to two conclusions:

  1. i) The diversity of conditions proves that the preparation of K2Ca(CO3)2, Fairchildite, is very delicate and difficult. We have tested the conditions of entries 3,6 and 7. K2Ca(CO3)2, Fairchildite was not obtained. The conditions of entry 3 led to a mixture of CaO and K2CO3, whereas the conditions of entries 7 and 8 led to a mixture of K2CO3 and CaCO
  2. ii) It is clear that the thermal treatment of vegetable plants is more efficient and easier.

To test the idea suggested by the reviewer, we studied the activity of the mixtures K2CO3, CaO., and K2CO3 and CaCO3, which derived from the precented thermal treatment. Yields were quantitative in each case. These results have been added to the table 1.

References about the preparation of (K2Ca(CO3)2).

  1. Arefiev, A.; Podborodnikov, I.; Shatskiy, A.; Litasov, K. Synthesis and Raman Spectra of K-Ca Double Carbonates: K2Ca(CO3)2 Bütschliite, Fairchildite, and K2Ca2(CO3)3 at 1 ATM. Геохимия 2019, 64, 967–973, doi:10.31857/S0016-7525649967-973.
  2. Pabst, A. Synthesis, Properties, and Structure of K2Ca(CO3)2, Buetschliit. Am. Mineral. 1974, 59, 353–358.
  3. Kröger, K. W. Illner, and W. Graeser, “Über die système alkalioxyd CaO–Al2O3–SiO2–CO2. XI. Die reaktionsdrucke im system K2O–CaO–SiO2–CO2,” Zeitschrift für Anorganische Und Allgemeine Chemie 1943, 251(3), 270–284.

Finally, the authors claimed that the Eco-base could be recovered and reused. However, the drying process includes heating at 500ºC. How are they sure that the base is really reused and that the drying process is not a re-activation process?

Can the authors provide a comparative experiment using the base after a regular drying (evaporation of ethyl acetate and 100ºC heating)?

We thank the reviewer for this suggestion. After new experiment, we can confirm that a drying of Eco-base at 100°C was sufficient for recycling the catalyst.  The heating at 500°C was not necessary. The condition of recycling has been modified.

After this revision, I will be happy to support the publication of this paper in Molecules.

Reviewer 2 Report

This is a very interesting and qualitative work. Presented investigation is an excellent example of an original approach to an already known problem. All of the research methods are competently applied and well described in the main text of the manuscript.

However, in my opinion, it is necessary to expand the section devoted to harvest and preparation of the Eco-bases. The authors should indicate how many samples of each plant culture were taken in total. What is the effect of a particular sampling location on the composition of the Eco-bases? Is there a difference when sampling close to the Zn smelter of the Bolesław Mine and Metallurgical Plant or some distance away from it? In addition, how much Eco-base is contained in each culture?

Some minor remarks:

The phrase "Error! Reference source not found." should be removed from the text. 

On the line 214 "Eco-catalysts" is repeated twice.

Once again, I would like to point out, this work is of very high quality and extremely interesting. This paper is recommended for publication after correcting the comments.

Author Response

This is a very interesting and qualitative work. Presented investigation is an excellent example of an original approach to an already known problem. All of the research methods are competently applied and well described in the main text of the manuscript.

However, in my opinion, it is necessary to expand the section devoted to harvest and preparation of the Eco-bases. The authors should indicate how many samples of each plant culture were taken in total. What is the effect of a particular sampling location on the composition of the Eco-bases? Is there a difference when sampling close to the Zn smelter of the Bolesław Mine and Metallurgical Plant or some distance away from it? In addition, how much Eco-base is contained in each culture?

We thank the Reviewer for this comment and agree that more detailed information on harvest and preparation of Eco-bases will be beneficial for the reader. Our sampling consisted of ~1000 individual plants per Zn-hyperaccumulator species, ~500 shrubs for the invasive plant species, and ~4 trees of S. alba.

Different precisions have been added about the origin and the using of these plants and the loss of weight form crop to thermal treatment in the “general procedures” part and the preparation of Eco-base to extend the section devoted to harvest and preparation of the Eco-bases:

2.2.1. Harvest and preparation of the Eco-bases

Arabidopsis halleri (Ah) leaves have been collected in heavily industrialized region of Poland in the vicinity of the Zn smelter of the Bolesław Mine and Metallurgical Plant near the city of Olkusz.

Nocceae caerulescens (Nc) and Anhyllis vulneraria (Av) leaves were harvested before flowering in Les Avinieres, a former Zinc mining site in Saint-Laurent-le-Minier.

Arabidopsis halleri (Ah), Nocceae caerulescens (Nc) and Anhyllis vulneraria (Av) are well-studied hyperaccumulators of Zn, which were widely used in phytoremediation program. The population from the investigated location was previously shown to contain extremely high and consistent concentrations of Zn in aerial parts (ref 21).

Fallopia japonica (Fj) is another alien invasive plant in European wetlands. Its has been collected in Saint-Bauzille de Putois and Thoiras, Gard, France, in a context of integrated wetland management by ETPB Gardons and Grison team. This is a highly invasive and widespread alien species in Europe, also listed as one of the most potent global invasive plant species.

Salix alba (Sa) leaves were harvested in Laon, Grand-Est region, France. Salix alba is a common tree species, which is native to the wetlands of the northern hemisphere, especially Europe, temperate Asia and North Africa. Its biomass is very abundant.

Oyster shells were given by oyster aqua-culturists from Thau pond in the Mediterranean area, France.

Leaves were first washed twice in deionized water, oven-dried at 80 °C. The obtained solid (30g) then thermally treated in an oven under air flow. The temperature program was consisted of a first gradient from 25 to 350 °C in 1 h then 2 h at 350 °C, followed by a second gradient from 350 to 550 °C in 1 h then 4 h at 550 °C, before a slow cooling in the oven to obtain Eco-base. (5g).

3.2.1 Preparation of Eco-bases

For Eco-bases from Zn-hyperaccumulators and the invasive species, bulk samples of leaves were collected from ~ 1000 individual plants within a sample area of ~0.1 km2. Sampling at the metalliferous sites was conducted across an area that is consistently affected by the source of pollution and the harvested plant biomass was thoroughly homogenized to minimize biases from site heterogeneity. As for the reference plant material S. alba, leaves were collected from 4 trees, from an area of 2 km2.

The dried leaves, from each of each plant species, were thermally treated under air flow at 550 °C for 4 h, in order to remove the organic matter.

Concerning the amount of available biomass, two cases can be considered:

  • The case of Zn hyperaccumulators, caerulescens, A. vulneraria: cultures and crops have been described in ref. 20. (“Assuming 70% ground cover (Robinson et al. 2003), the number of A. vulneraria plant per hectare was estimated at 266 160 in the site of Les Avinières. The theoretical yield per hectare was 1791 kg.ha-1 of dry biomass. This result has been compared with the Zn-hyperaccumaltor reference, N. caerulescens. The theoretical yield per hectare was 1632 kg.ha-1). Concerning A. halleri, the theoretical yield per hectare and the size of leaves were comparable to N. caerulescens.

  • The case of non-accumulator (F. japonica): this plants species is an invasive alien species on the entire European continent (see page 5, line 170). The resource is huge, even unlimited.

Finally, in this work, ZnO is not the active species. Thus, Eco-bases which derived from Polish or French Zn hyperaccumulators (A. hallleri, N. caerulescens, A. vulneraria) or from non-accumulator (F. japonica) are the same reactivity. As indicated, the key point is the natural richest in potassium of all these species.

Some minor remarks:

The phrase "Error! Reference source not found." should be removed from the text. 

Indeed, there was a problem of pdf conversion with the references to Tables. It has been corrected.

On the line 214 "Eco-catalysts" is repeated twice.

The correction has been made.

Once again, I would like to point out, this work is of very high quality and extremely interesting. This paper is recommended for publication after correcting the comments.

Reviewer 3 Report

See attached file.

Author Response

The manuscript entitled “Mechanochemistry and Eco-bases for Sustainable Michael Addition Reactions“ written by Yvette Lock Toy Ki, Armelle Garcia, Franck Pelissier, Tomasz Olszewski, Alicja Babst-Kostecka and Claude Grison presents an environmental friendly method for Michael additions. The manuscript is well-written and readily understandable. The following questions and issues arose after reading:

Page 2, row 72: „..in numerous mechanical-assisted Michael addition reactions.” Instead “..some mechanical-assisted Michael addition reactions.” should be written.

The correction has been made

Page 3, row 91: “chloromorm” should be replaced by “chloroform”

The correction has been made

In paragraph 2.2.2. Past tense should be used.

The corrections have been made

Page 3, row 163: “The crude is filtered.” Did you mean the suspension was filtered, and the filtrate was concentrated? Please clarify.

The authors agree with the need of clarification. The sentence has been modified.

The suspension was filtered, and the filtrate was concentrated. The yield was obtained by using GC-FID with biphenyl as internal standard. »

Page 4, row 181; page 5, row 216; page 6, row 236; page 7, row 248; page 8, row 298; page 10, row 338: References are missing (Error! Reference source not found.)

Indeed, there was a problem of pdf conversion with the references to Tables. It has been corrected.

Page 4, Table 1: CaO and eco-CaO gave the same excellent conversion, but the yields were different. What was the reason of this? Result of using KHCO3 in the reaction of cyclopent-2-en-1-one is missing (but referred in the text). CaO and eco-CaO were not used in the reaction of 2-pentyl-cyclopent-2-en-1-one. Why?

CaO and Eco-CaO led to a different yield, because the composition and the reactivity of these Eco-bases is different. With Eco-CaO, the mineral composition included Ca and other metallic elements (Na,K, Mg, Fe, Zn, Mn; see ref 12), which seemed less efficient.

Result of using KHCO3 in the reaction of cyclopent-2-en-1-one has been added in the Table.

CaO and Eco-CaO were not used in the reaction of 2-pentyl-cyclopent-2-en-1-one, because their preparation required high temperature (>900°C). It was more advantageous to work with carbonates. Their life cycle assessment is better.

Page 5, row 187; page 9, row 305, and page 10, row 344: “Yields were determined after distillation (of the crude mixture.)” This is not described in the experimental part. In the supporting information, it is written that the solvent was evaporated. Please describe and unify the details.

We agree with this remark. Details about the distillation or the purification by chromatography (chalcone) have been added in the experimental part.

Page 5, row 196: “exihited “should be “exhibited”

Page 7, row 253: “KSO4” should be “K2SO4”

Page 8, row 297: “2-alkyl-cyclopente-2-enones” is written, but only 2-pentylcyclopent-2-en-1-one was used. Plural is inadequate.

The corrections have been made.

Page 9, Table 4 entries 2-6: Why did the yields differ, when the conversion was the same?

Page 9, row 314: There is no entry 11 in Table 4.

Page 10, rows 326-330: Font size is different.

The corrections have been made.

Page 10, Table 6, entry 3: Using chalcone, conversion was 91%, but yield was only 61%. Were there any side products / byproducts? Why only ketones were used as Michael acceptors? Nitrostyrene, cinnamonitrile and methyl cinnamate should be tested to show the extensibility of the method.

Ketones were used as Michael acceptors, because they are the best substrates.

We tested nitrostyrene, cinnamonitrile and methyl cinnamate. 38% yield was obtained with nitrostyrene. The reaction failed with cinnamonitrile (0%) and methyl cinnamate (5%).

New attempt was carried out with butanone. Conversion was total, but a partial polycondensation of adduct was observed. The expected product was obtained with a yield of 61%. The screening of Michael acceptors revealed that the methodology was especially well adapted to enones.

These results and conclusions have been added in the main text.

Page 11, row 357: “2-cyclo-hexen1-one” should be “cyclohex-2-en-1-one”

The correction has been made.

Page 11, row 359: “.chalcone, the less reactive Michael acceptor.” Please explain this.

Chalcone and its related compounds are known to be π-conjugated compounds, which are less reactive that non-aromatic enones.

Page 11, row 363: “cyclopentanedione” should be “cyclohexane-1,3-dione”

The correction has been made.

The reviewer recommends accepting the manuscript after further experiments and answering questions.

Round 2

Reviewer 1 Report

The authors followed all my former concerns. The manuscript was improved, so I support its publication in its current form.

Reviewer 3 Report

The authors answered all questions and performed the suggested experiments.

The reviewer recommends acceptance of the manuscript after a thorough reading. There are some typos (e.g. "gaz" instead of "gas").